# Human Perception of the Emotional Expressions of Humanoid Robot Body Movements: Evidence from Survey and Eye-Tracking Measurements

**DOI:** 10.3390/biomimetics9110684

**Published:** 2024-11-08

**Authors:** Wa Gao, Shiyi Shen, Yang Ji, Yuan Tian

**Affiliations:** 1Co-Innovation Center of Efficient Processing and Utilization of Forest Resources, Nanjing Forestry University, Nanjing 210038, China; 2College of Furnishings and Industrial Design, Nanjing Forestry University, Nanjing 210038, China; shenshiyi5324@163.com (S.S.); jiyang@njfu.edu.cn (Y.J.); t1498363838@163.com (Y.T.)

**Keywords:** emotional expression, human perception, human–robot interaction, robot body movements

## Abstract

The emotional expression of body movement, which is an aspect of emotional communication between humans, has not been considered enough in the field of human–robot interactions (HRIs). This paper explores human perceptions of the emotional expressions of humanoid robot body movements to study the emotional design of the bodily expressions of robots and the characteristics of the human perception of these emotional body movements. Six categories of emotional behaviors, including happiness, anger, sadness, surprise, fear, and disgust, were designed by imitating human emotional body movements, and they were implemented on a Yanshee robot. A total of 135 participants were recruited for questionnaires and eye-tracking measurements. Statistical methods, including K-means clustering, repeated analysis of variance (ANOVA), Friedman’s ANOVA, and Spearman’s correlation test, were used to analyze the data. According to the statistical results of emotional categories, intensities, and arousals perceived by humans, a guide to grading the designed robot’s bodily expressions of emotion is created. By combining this guide with certain objective analyses, such as fixation and trajectory of eye movements, the characteristics of human perception, including the perceived differences between happiness and negative emotions and the trends of eye movements for different emotional categories, are described. This study not only illustrates subjective and objective evidence that humans can perceive robot bodily expressions of emotions through only vision but also provides helpful guidance for designing appropriate emotional bodily expressions in HRIs.

## 1. Introduction

Body movements are not only ways for humans to express their emotions but are also ways to help perceive the emotional states of others in social interactions. Humans can discern various affective states expressed by body movements [1], and the recognition accuracy is comparable to that of facial expressions, regardless of the static and dynamic conditions present [2,3]. Body movements are a feasible approach for perceiving emotions at long distances [4]. In recent years, humanoid robots have been widely used in healthcare [5,6,7,8], guidance [9], education [10,11], public places [12,13,14,15], and other scenarios. Because of their human-like appearances, they are expected to have the ability to express emotions like humans. Robots with human-like emotional abilities are perceived as more likeable, leading to more pleasurable interactions and better mutual understandings [16,17]. Currently, studies of emotional expressions in HRIs from the point of view of body movements are gradually increasing. 

To effectively establish emotional interactions based on body movements in HRIs, two aspects are necessary to explore. One aspect is determining how to express emotion via humanoid robot body movements, and the other factor is the corresponding perception of humans. There have been studies on implementing emotional body movements into various humanoid robots, such as Pepper, Nao, Kobian, Roman, Greta, and Max [3,18,19,20,21,22,23]. For example, in the study of Tsiourti et al., body animations using the head, torso, and arms were used to generate the robot’s emotional expressions [3]. Erden explored emotional postures for three emotions—anger, sadness, and happiness—using the humanoid robot Nao [18]. Hsieh et al. presented an analysis of the impressions resulting from the implementation of human behaviors into Pepper the robot, and they designed different body movements to express cute and cool greeting styles [19]. De Silva et al. found that vertical features and features indicating the lateral opening of the body are informative for differentiating happiness and sadness [20]. Takahashi et al. designed different body movements to express five emotions, including joy, anger, shame, small sadness, and large sadness, in the robot Nao [22]. Some studies also focused on human emotional perceptions toward robots [24,25,26]. For instance, the work of Hwang et al. revealed that the robot’s appearance can arouse emotions such as concern, enjoyment, and favor, and different designs of the head, limb, and body of the robot generate different perceived personalities [25]. Ghafurian proposed design guidelines for 11 affective expressions for the Miro robot and further studied how the robot was perceived in general [26]. However, there are relatively few studies in the field of the human perception of humanoid robot emotional body movements [27,28]. 

For this purpose, the robot’s body movements and expressed emotions need to be designed. Then, the corresponding perceptions of humans can be evaluated and analyzed on this basis in this paper. Note that the implementation of body movements into a robot is constrained by its physical structure. The emotional body movements mentioned in relevant studies are designed based on humanoid robots with more degrees of freedom (DoF). It is difficult to determine the portability of these movements in our robot platform, as it only has 17 DoF. Hence, we directly consider the imitation of human emotional motions. Although no robot really moves as a human does [29], some studies have shown that robots can express emotions by imitating human movements [30,31,32,33]. The emotional databases of human motion can prove beneficial for designing the emotional body movements of robots. Human perceptions of HRIs are usually evaluated by investigations, questionnaires, eye tracking, electrocardiograms (ECGs), heart rate variability (HRV) measurements, electrodermal activity (EDA) measurements, and so on [19,28,33,34]. In our study, we implement the evaluation from both subjective and objective viewpoints. As mentioned above, the details of the study in this paper are shown as follows.

Design the robot’s body movements from the viewpoint of imitating human emotional movements to express different emotions that can be perceived by humans and provide a guide to emotional HRI design for humanoid robots.Explore the human perceptions of different emotional body movements of robots, including perceptions of emotional category, intensity, and arousal, to provide references for the design of emotional body movements in HRIs.

The structure of this paper is shown as follows. Section 2 briefly illustrates the related works, including the bodily expressions of emotions and the evaluation of human emotional perception in HRIs. Section 3 shows the employed robot platform, the designed robot’s emotional body movements, and the implemented procedures of subjective and objective measurements. Section 4 analyzes the self-reporting data obtained by subjective measurements and the eye-tracking data obtained by objective measurements. Section 5 discusses the emotional expressions of the designed robot’s body movements and the corresponding perceptions of humans, including emotional intensity, recognition rate, arousal, and eye movements. Section 6 presents the conclusion and some limitations of this study.

## 2. Related Works

### 2.1. Bodily Expression of Emotion 

Bodily expressions play an indispensable role in human communication, as they can convey human emotions that may influence social relationships [32,35]. Previous reports on whole-body expressions can be traced back to the early work of Darwin, but extremely few stimuli sets were described in detail until around twelve years ago [36,37,38,39,40,41]. Atkinson et al. developed a set of dynamic and static whole-body expressions consisting of 10 identities displaying whole-body expressions (anger, disgust, fear, happiness, and sadness) in full-light and point-light displays [2]. De Gelder et al. constructed a database of 254 face-blurred whole-body expressions (anger, fear, happiness, and sadness) for a bodily expressive action stimulus test (BEAST) and discussed the ability to recognize these four emotions [37]. These works reveal a priori differences in the recognizability of bodily emotions [32,34,38]. Zhang et al. reported a human body kinematic dataset and a multi-view emotional expressions dataset (MEED) using 2D pose estimation [36,40]. Both datasets present six emotional (anger, disgust, fear, happiness, sadness, and surprise) and neutral body movements. In recent years, the study of body movements has gradually shifted from conceptual to data-based quantitative research [42]. The studies can be used and applied to better design emotional body movements for humanoid robots.

There have been some studies regarding the generation of robot emotional body movements based on actual human behaviors. Erden described human behaviors in qualitative terms and encoded them in quantitative terms to generate robot emotional body movements associated with six basic emotions (anger, disgust, fear, happiness, sadness, and surprise) [18]. In his study, some emotional postures of the Nao robot are generated based on Coulson’s quantitative descriptions of postures described with simplified human body models [18,42]. De Silva et al. and De Meijer et al. found that features and trunk movements (ranging from stretching to bowing) can be used to distinguish positive and negative emotions [20,43], and Tsiourti et al. designed a robot’s body movements to express happiness, sadness, and surprise based on these correlations [3,44]. McColl et al. developed emotional body language for a human-like social robot, Brian 2.0, expressing eight emotions (sadness, elated joy, anger, interest, fear, surprise, boredom, and happiness) through a variety of body postures and movements identified in human emotion research [45]. Mizumaru et al. implemented four emotional movements (fear, pride, happiness, and sadness) in the Nao robot based on the key poses proposed by Beck [46,47]. Yagi et al. revealed that the motion of vertical oscillation affects the human perception of robots through three gait-like motions expressing anger, happiness, or sadness in the Ibuku robot [48,49]. In Guo et al.’s work, five types of emotional behaviors—joy, fear, neutral, sadness, and anger—were designed and presented to users using humanoid Alpha 2 robot with 20 DoFs [28]. The existing research mentioned above indicates that a robot’s body movements can be designed to express different emotions according to human behaviors. It also reveals the importance of human emotional body movements for the design of robot bodily emotional expression. 

### 2.2. Evaluation of Human Emotional Perception in HRIs

Whether the emotional body movements of robots can be perceived is another important aspect for researchers; relatively few studies have focused on this compared to robots’ bodily expression of emotion. Subjective methods such as questionnaires, Likert scales, and self-reports are the most common methods to evaluate the perception of human emotions [28,50,51]. For example, in the works of Erden and Tsiourti et al., the evaluations were implemented by asking participants to watch different videos and choose one of the listed emotions as the most representative for each video [3,18]. In the experiments designed by McColl et al., participants were required to watch two separate videos, first of a robot and then of a human actor displaying emotional body language, and to make a choice from a list of eight possible emotions [45]. Fernández-Rodicio et al. recruited participants to watch interaction videos and asked them to select a value between 1 and 10 for different descriptors to find relationships between the amplitude and speed of the robot’s expressions and the mood perceived by the users [50]. Kaushik et al. generated body movements using the Quori robot and evaluated how much each movement displayed the eight different emotions by watching videos and using a five-point Likert scale [51]. Few studies have explored humans’ emotional perception of the robot movements using objective methods such as measurements of various physiological signals. Hsieh et al.‘s work adopted a scale for scoring the expressions of robots and measured the HRVs of users to study the relationships between robot expression patterns and human preferences [19]. Guo et al. provided a multimodal method for evaluating users’ emotional responses that included subjective reporting, pupillometry, and ECG and found that a humanoid robot’s emotional behaviors could evoke significant emotional responses in users [28]. These studies show that it is feasible to explore humans’ perceptions of the emotional movements of robots from both subjective and objective perspectives. 

## 3. Methods

### 3.1. Robot Platform

A humanoid robot named Yanshee, with a height of 37 cm, was employed. This robot had a total of seventeen DoFs, named D1 to D17, including one in the head, six in the arms and ten in the legs, as shown in Figure 1. The Yanshee robot utilized the open hardware platform architecture Raspberry Pi+STM32 and could achieve interactions through multiple means, such as through its voice, by walking, through arm actions, and through sound.

### 3.2. Materials of Robot’s Emotional Body Movements 

The robot’s emotional body movements were designed by considering and imitating the works of Wallbott, Tsiourti et al., De Meijer et al., and Zhang et al. [3,4,36,43]. For example, happiness leads to the robot waving its hands and fear leads to it crouching down. The emotional body movements were expressed by the robot’s head, arms, and legs and evaluated through consensus amongst the researchers. A total of 19 behaviors labeled B1–B19 were designed, as shown in Figure 2, and the details are listed in Table 1. B1~B5 were designed to express happiness, B6~B8 to express anger, B9~B11 to express sadness, B12 and B13 to express surprise, B14~B16 to express fear, and B17~B19 to express disgust. The six emotions of happiness, anger, sadness, surprise, fear, and disgust were labeled E1, E2, E3, E4, E5, and E6, respectively. DoFs D5 and D14, located in the robot’s hip, were not used in the design process. 

### 3.3. Measures 

Both subjective and objective methods were adopted to verify whether behaviors B1~B19 could express corresponding emotions and to explore whether they could be perceived by humans. For subjective data, questionnaires were used to obtain self-reports from humans for different emotional categories and intensities. For objective data, eye tracking was used to analyze pupil diameters, saccade counts, fixation duration, and trajectory. Larger pupillary changes corresponded to stronger emotional arousal levels [52,53,54], and saccade counts, fixation durations, and trajectories were considered useful for determining the characteristics of user perceptions [55,56,57]. 

#### 3.3.1. Implementation of Questionnaire 

A total of 66 participants with the age range of 18–30 years were recruited from the same university to complete the questionnaires, of which there were 33 males (M = 23.48, SD = 1.62) and 33 females (M = 23.39, SD = 1.09). They volunteered to participate after being fully informed of the procedures. The participants were required to watch videos of the 19 behaviors shown in Figure 2 and Table 1. The order of the videos was random. The participants had a few seconds to rest after watching different videos. After each video, they were asked to answer six questions to score the emotion intensities for the behaviors representing the six emotions of happiness, anger, sadness, surprise, fear, and disgust. A 7-point Likert scale, in which 7 corresponded to “strong” and 1 corresponded to “weak”, was used here. 

#### 3.3.2. Eye-Tracking Measurement

Eye-tracking measurement was implemented using the Ergo LAB human–machine environment synchronization cloud platform and the Tobii Pro Fusion eye-tracking instrument, which were both developed by KINGFAR International Inc. The sampling rate was set to 250 Hz. Another 69 students were recruited from the same university, of which there were 34 males (M = 22.44, SD = 2.26) and 35 females (M = 22. 63, SD = 2.07). All participants had normal or corrected-to-normal vision without any other eye disease and were self-reported to be free from any history of neurological or psychiatric disorders. They participated after being fully informed of the experimental procedures and were paid. 

Five areas of interest (AOIs), the head, hands, torso, legs, and feet, were set in the eye-tracking measurement system, as shown in Figure 3a. The participants were required to sit in front of the screen and watch 19 videos. The employed eye-tracking system set the videos to play in random order. After each video, a black screen served as the resting page for a few seconds. Figure 3b shows a scenario in which a participant was watching a video of the robot’s body movement. 

## 4. Analysis

### 4.1. Analysis of Questionnaires

The questionnaire data were analyzed to verify whether the designed emotional behaviors belonged to the prearranged emotional category and to identify the corresponding intensities. In addition, significant differences in robot behaviors were compared. 

The data showed a normal distribution, and the mean values (M-values) of participants’ self-reports for each behavior in different emotional categories are illustrated in Figure 4. The recognition rate was calculated first. The average value 3.5 was chosen as the threshold of intensity. When the M-value was larger than the threshold, the intensity of the emotional body movement could be perceived more easily. It was found that the emotional category that the robot’s body movement belonged to was slightly different from our original design. B3, B10 and B18 showed quite low recognition rates and low emotional intensities that were not suitable for emotional bodily expressions. On this basis, the data of adjusted emotional categories based on participants’ self-reports were classified into four levels from strong to weak, named L1 to L4, using k-means clustering, as shown in Table 2. The corresponding emotional recognition rates were also given. The robot’s behaviors shown in Table 2 from top to bottom were ranked in the order of the intensity of the M-value from high to low. Table 2 presents how the different emotional categories corresponded to each other; it showed four levels of happiness and fear and two levels of anger, sadness, and disgust. Considering an average intensity level could be needed for reference in some scenarios; another classification established using k-means clustering and consisting of three levels is also given in Appendix A. The three levels from strong to weak were named TL1 to TL3, as shown in Table A1. 

The repeated ANOVA test was employed to analyze significant differences in human perceptions of the robot’s different emotional behaviors in the same emotional category. According to the results of Mauchly’s test of sphericity and tests of within subject effects, there were significant differences in the perception of robot behaviors for the emotional categories of happiness (E1), anger (E2), sadness (E3), and fear (E5), while there were no differences for surprise (E4) and disgust (E6). For happiness, *F*(2.965,192.706) = 36.298, *p* < 0.001, and partial *η*^2^ = 0.358. For anger, *F*(1.96, 127.42) = 8.088, *p* < 0.001, and partial *η*^2^ = 0.11. For sadness, *F*(2.631, 171.014) = 8.088, *p* = 0.011, and partial *η*^2^ = 0.058. For surprise, *F*(1, 65) = 0.024, *p* = 0.878, and partial *η*^2^ < 0.001. For fear, *F*(2.987, 194.187) = 30.029, *p* < 0.001, and partial *η*^2^ = 0.316. For disgust, *F*(1.984, 128.974) = 1.746, *p* = 0.179, and partial *η*^2^ < 0.026. The significance levels between different robot behaviors in each emotional category could be easily found using pairwise comparisons, as shown in Figure 5. For happiness, there was no significant difference between B2 and B5. For anger, there was no significant difference between B6 and B8. For sadness, B14 did not show significant differences between B9 and B11 or between B11 and B16. For fear, B15 showed no significant differences between B14 and B16.

### 4.2. Eye-Tracking Analysis

The pupil diameter data showed a normal distribution according to the Kolmogorov–Smirnov test, but the saccade count data did not pass the test with *p* < 0.05. The proportions of total fixation duration for the Yanshee robot and the proportions of first fixation duration for different AOIs were considered, and the corresponding data were not normally distributed. For readability, the sequence numbers of the robot’s behaviors in the following sections were listed in the following order. The maximum recognition rate of that behavior is shown in Table 2. 

#### 4.2.1. Analysis of Pupil Diameter and Saccade Count 

Repeated ANOVA was employed to analyze the pupil diameter data, and Mauchly’s test of sphericity was also utilized. The average pupil diameters showed significant differences for the robot behaviors in the categories of happiness, anger, and fear, respectively, with *F*(3.143, 213.747) = 6.486, *p* < 0.001, partial *η*^2^ = 0.087; *F*(1.838, 124.992) = 4.973, *p* < 0.05, partial *η*^2^ = 0.068; and *F*(3.503, 238.226) = 2.957, *p* < 0.05, partial *η*^2^ = 0.042, while the maximum pupil diameters did not show these differences. Both the average pupil diameters and the maximum pupil diameters showed significant differences in the category of surprise with *F*(1, 68) = 7.582, *p* < 0.01, partial *η*^2^ = 0.1 and *F*(1, 68) = 9.051, *p* < 0.01, partial *η*^2^ = 0.117, respectively. For the categories of sadness and disgust, there were no significant differences in the average pupil diameters and the maximum pupil diameters. The significance levels of the average pupil diameters regarding the robot’s different behaviors in the categories of happiness, anger, surprise, and fear could be found through pairwise comparisons, as shown in Figure 6a–d. The significance levels of the maximum pupil diameters for surprise are shown in Figure 6e. The corresponding estimated marginal means are also shown in Figure 6. Friedman’s ANOVA was employed to analyze the saccade count data. There were significant differences in the saccade counts for each emotional category of robot behaviors. The estimated marginal means of saccade counts for the robot body movements are shown in Figure 7a. 

#### 4.2.2. Analysis of Fixation Duration and Trajectory

Friedman’s ANOVA was employed to analyze the proportions of fixation duration. For the first fixation duration of each AOI when the Yanshee robot expressed different behaviors, the results revealed that there were significant differences in the same emotional category and that participants could spot the parts of the robot that were moving. The behavior of happiness B1 was taken as an example. The body movements expressing happiness were implemented by hands. It was found that the robot’s hands accounted for the largest proportion of the first fixation duration (10.05%), while the other four AOIs, head, torso, legs, and feet, accounted for 4.25%, 7.55%, 3.08%, and 0.49% of the total, respectively. The AOIs with the highest proportions of the first fixation duration for the robot’s different behaviors are shown in Figure 7b. 

The torso comprised the largest proportion of the total fixation duration. The corresponding heat maps are shown in Figure 8, in which red represents the longest fixation duration, followed by yellow and green. The trajectories of the participants’ eye movements are shown in Figure 9. 

## 5. Discussion

### 5.1. Perceptions of Emotional Category, Intensity, and Arousal

Some of the body movements of the robot are confused for different emotions, as shown in Figure 1 and Table 2. The intensities and recognition rates of these confused behaviors differ in different emotional categories. For example, B12 is recognized as both happiness and surprise, with recognition rates of 63.64% and 36.36%, respectively. B7 is perceived as anger and disgust, with recognition rates of 65.15% and 31.82%, respectively. B15 is recognized as fear and disgust, with recognition rates of 81.82% and 18.18%, respectively. B9, B14, and B16 are all seen as sadness and fear. This indicates that there is some confusion in the human perception of robot emotional movements, which is consistent with the work of de Gelder et al. concerning confusion in the perception of human emotional movements [37]. The body movements designed for happiness and fear in this paper show relatively high recognition rates. This means that, with proper design, even though robots have relatively few DoFs, they can express easily recognized positive and negative emotions through their body movements. Using more DoFs in the design of a robot’s body movements does not correspond to better emotional expressions. Compared to the recognition rates in other emotional categories, it is found that the recognition of happiness is usually superior to the recognition of negative emotions, including sadness, anger, and disgust. This is consistent with the studies of Bandyopadhyay et al. and Goeleven et al. about human facial emotion recognition [56,57] and with the work of de Gelder et al. about whole-body emotional expression [38]. 

Table 2 compares the intensities of different emotional behaviors. For example, when B4 and B16 are used to express happiness and fear, respectively, their intensities are found to be similar. Moreover, when combined with Figure 5, it is shown that different body movements can have different levels in the same emotional category. For example, there are significant differences in the intensities of B4 and B1 in the case of happiness, with the intensity of B4 being larger than B1. This means that humans can perceive the degree of happiness expressed by these two movements differently and sense more happiness in B4. In the case of fear, there is no significant difference between B15 and B14, but the recognition rates of the participants are quite different. This means that B15 can be recognized more easily than B14. These results are beneficial for many scenarios in which robots need to express different levels of multiple categories of emotions and provide a reference for making robot expressions more human-like.

Pupil diameter can be used to evaluate emotional arousal. The body movement with the greatest arousal is B6 (angry) (M = 3.98 mm, SD = 0.78 mm), followed by B4 (happy) (M = 3.97 mm, SD = 0.81 mm) and B15 (fear) (M = 3.96 mm, SD = 0.79 mm), according to Figure 6. There are no significant differences in arousal for the corresponding behaviors of anger and disgust. For happiness, the arousal order is B4, B1, B5, B2, and B12. For anger, it is B6, B7, and B8. For fear, the arousal order is B15, B16, B14, B9, and B19. B4 shows significant differences with B2 and B12, while B6 shows significant differences with B7 and B8, respectively. B15 has no significant differences with B16. Considering the data for happiness and anger in Table 2, B4 has high emotional intensity, recognition rate, and arousal, which means that it can be used when the robot needs to express a high level of happiness. And even though B6 results in high arousal, its recognition rate is quite weak. The human perception of B6 may not be better than B8 when the robot needs to express mild anger. Compared with B15, B16 has a higher emotional intensity and recognition rate, but there is no difference in arousal. This means that B15 and B16 can be used to express different levels of fear. Higher arousal does not indicate higher emotional intensity or recognition rate. A reasonable choice should be made based on the real needs of HRIs. 

### 5.2. Guide for Grading the Designed Robot’s Bodily Expressions of Emotion

The guide for grading the robot’s designed bodily expressions of emotion can be summarized according to the analysis of pupil diameter for different behaviors in Section 4.1, the details of movements shown in Table 1, and the levels shown in Table 2. The expression of happiness is easier to design, since it just focuses on the movement of hands to obtain expressions with higher recognition rates, different intensities, and different arousals. B4, B1, B12, and B5 can be used to express four levels of happiness from strong to weak, since B2, which has a relatively lower recognition rate, does not show significant differences in terms of emotional intensity and average pupil diameter with B5. However, it should be noted that B12 can be perceived as happiness mixed with a certain degree of surprise, as its recognition rate and the intensity of surprise are not very low. The behavior of the robot’s arm akimbo is a feasible way to express anger. B7 and B8 can be used to express two levels of mild anger, with no significant difference in arousal, while B7 can be perceived as anger mixed with disgust. B9 and B11 can be used to express two different kinds of sadness with no difference in arousal but a difference in intensity. B9 can be perceived as sadness mixed with a certain degree of fear. B11 shows a better recognition rate and lower intensity compared with B9. B13 can be used to express surprise to a certain extent, but it is better to optimize on this basis. B16, B15, B14, B9, and B19 can be used in different combinations to express different levels of fear. For example, the combination of B16, B14, B9, and B19 or the combination of B15, B9, and B19 can be used to express different levels of fear in a scenario since there are no differences between B15 and B16 in intensity and arousal, but there are differences between B16 and B14 in intensity. B15 can be perceived as fear mixed with some disgust, and B14 expresses fear mixed with sadness. B17 can be used to express mild disgust. 

### 5.3. Corresponding Characteristics of Human’s Perception 

The saccade counts, fixations, and trajectories of the participants’ eye movements show different characteristics when perceiving different emotional body movements of the robot. Higher saccade counts correspond to a longer searching process for required information. Participants show the highest saccade count for B15, which is followed by B13 and B14. Compared to these three samples, it is found that using more DoFs may not make the search process more complex. This may be because participants may not be concerned with changes in specific joints of the robot and will just focus on the overall impression. As shown in Figure 8 and Figure 9, the participants usually focus on the hands when the robot expresses happiness, and the trajectories of their eyes show a horizontal trend. However, for sadness and fear, which have high recognition rates, the fixate on the torso and legs. For anger, sadness, fear, and disgust, the eye movement trajectories are vertical. The correlations between the pupil diameters, saccade counts, and proportions of total fixation durations are analyzed using the Spearman correlation test. The results reveal that there are no correlations between the pupil diameters and proportions of fixation durations, and the correlations between the saccade counts and proportions of fixation durations are negative (*p* < 0.01). This reveals that the longer the total fixation duration is, the shorter the searching process for required information and the easier the perceptions of the robot’s body movements. 

The effects of gender are also considered. There are no significant differences in gender (*p* > 0.05) for the self-reports, pupil diameters, and saccade counts. However, there are significant differences for some behaviors in the proportions of the initial and total fixation durations. For example, when considering the proportions of the first fixation durations, males are more fixated on the torso for B7 and less fixated on the head for B8 and legs for B14 than females (*p* < 0.05). When considering the proportions of total fixation duration, males are less fixated on the head for B8 and B16 and the legs for B14 than females (*p* < 0.05). However, areas with different fixations are not the main moving parts of the robot. Hence, gender may not influence human perceptions of robot emotional body movements. 

## 6. Conclusions

This study explores the design of emotional body movements for a humanoid robot and the human perceptions of them. Nineteen behaviors belonging to six emotional categories, namely, happiness, anger, sadness, surprise, fear and disgust, were designed by imitating human bodily expression of emotion with the Yanshee robot. Self-reporting and eye-tracking measurements were employed, and the corresponding data were analyzed using statistical methods to evaluate the designed body movements of the robot and the ability of humans to perceive these movements. The main contributions of this study are as follows:It provides subjective and objective evidence that humans can perceive robots’ emotional movements through vision alone, which provides a reference for designing long-distance human–robot emotional interactions.It proposes a guide for grading the designed robot’s bodily expressions of emotion, which can help with the delicate expression of the emotions of robots in real-world environments. Designers can adopt different emotional expressions for robots according to their needs in real-world environments.It summarizes the corresponding characteristics of human perception, which is helpful for regulating human emotions in HRI processes. Humans have a greater ability to perceive happiness than negative emotions, such as fear, sadness, anger, and disgust. Eye movements have different characteristics when perceiving happiness and negative emotions. Gender may not affect human perception of emotional body movements for robots.It also showed that robots with relatively few DoFs can express strong and diverse emotions with proper body movement design, and humans have the ability to perceive these emotions.

However, limitations also exist for this study, which can be summarized in three aspects. First, the participant groups recruited in the experiments were young. The results cannot represent all age groups. Second, only eye-tracking measurements were employed in the objective analysis. It may be better to use multimodal measurements, such as ECG and HRV, to verify the results from multiple perspectives. However, we do not have access to such technical conditions at this stage. Third, the participants in this study were all Chinese. It is not certain whether the results would be the same in a different country or culture. Future work will focus on the applications of robot bodily expressions in different environments and explore corresponding design strategies for human emotional regulation in HRIs.

## Figures and Tables

**Figure 1 biomimetics-09-00684-f001:**
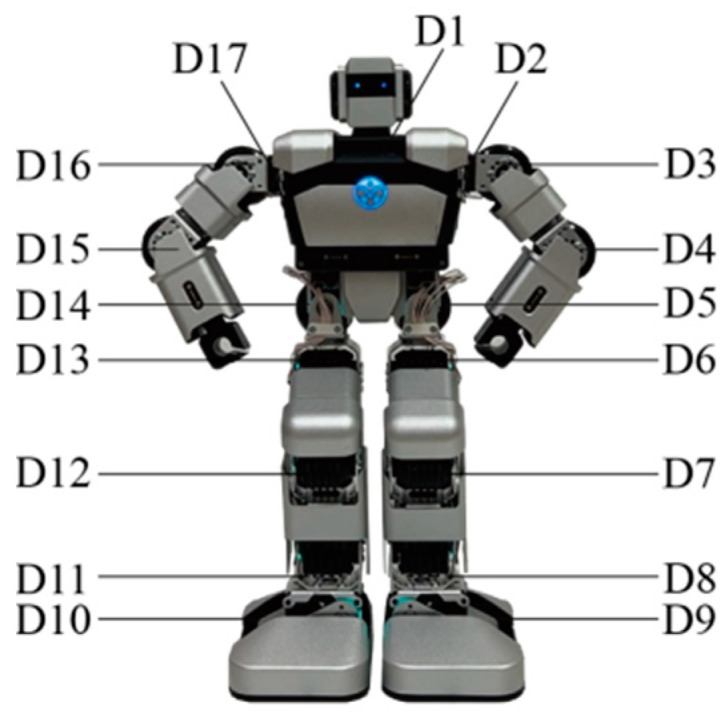
Yanshee robot.

**Figure 2 biomimetics-09-00684-f002:**
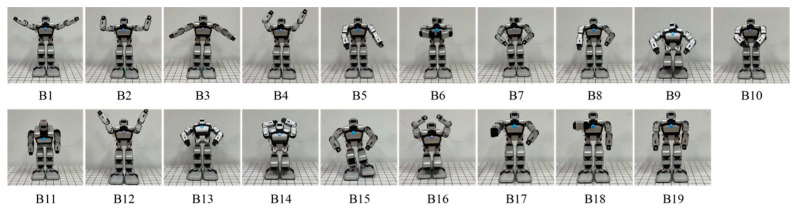
Designed robot’s emotional body movements.

**Figure 3 biomimetics-09-00684-f003:**
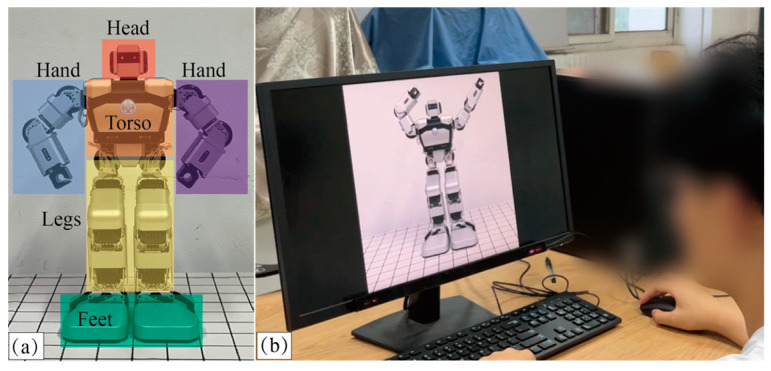
(**a**) Five AOIs of Yanshee robot. (**b**) Experimental scenario.

**Figure 4 biomimetics-09-00684-f004:**
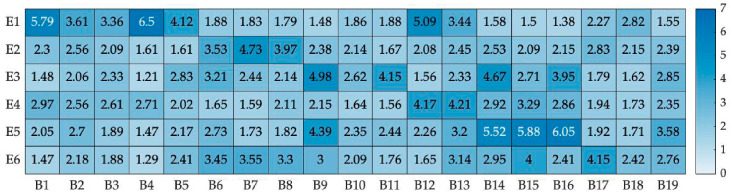
M-values of self-reports obtained by questionnaires.

**Figure 5 biomimetics-09-00684-f005:**
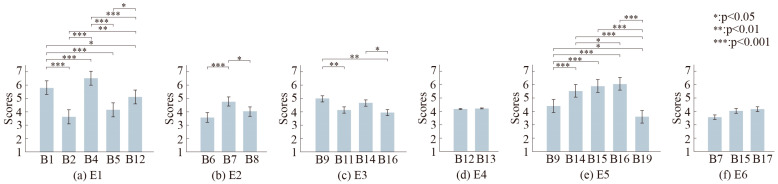
Estimated marginal means of data obtained by participants’ self-reports in different emotional categories. (**a**–**f**) represent the results obtained from the adjusted emotional categories. The error bars in the legend represent the 95% confidence interval.

**Figure 6 biomimetics-09-00684-f006:**
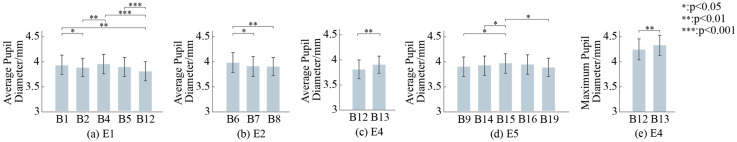
(**a**–**d**) Estimated marginal means of average pupil diameters in the categories of happiness, anger, surprise, and fear, respectively. (**e**) Estimated marginal means of maximum pupil diameters for surprise. The error bars in the legend represent the 95% confidence interval.

**Figure 7 biomimetics-09-00684-f007:**
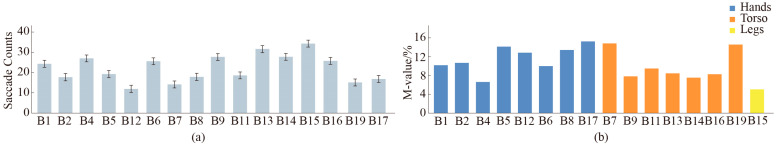
(**a**) Estimated marginal means of saccade counts. The error bars in the legend represent the 95% confidence interval. (**b**) AOIs with the highest first fixation durations.

**Figure 8 biomimetics-09-00684-f008:**
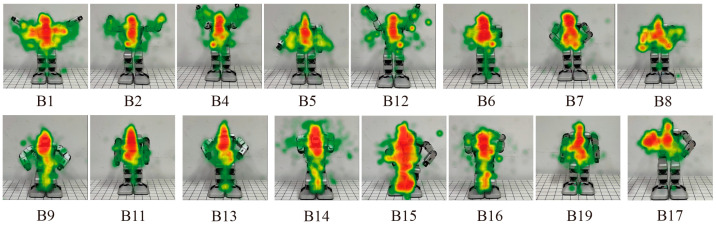
Heat maps for designed behaviors shown in Table 2.

**Figure 9 biomimetics-09-00684-f009:**
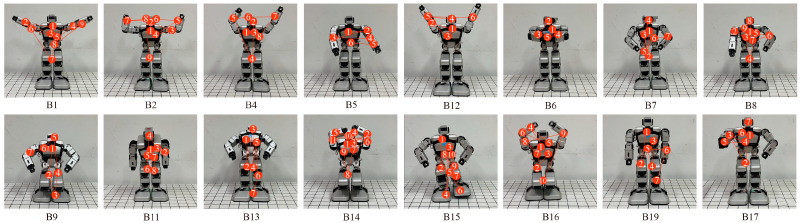
Trajectories of participants’ eye movement.

**Table 1 biomimetics-09-00684-t001:** Details for designed emotional body movements.

Emotion	Behaviors *	DoF *	Description
E1 (Happiness)	B1	D3, D4, D15, D16	Raise arms
	B2		Raise arms and make forearm vertically
	B3		Stretch arms laterally
	B4		Raise and wave arms
	B5	D2, D3, D4, D15, D16, D17	Wave arms in front of body
E2 (Anger)	B6	D1, D2, D3, D4, D15, D16, D17	Hold arms together and turn the head sideways
	B7	D1, D3, D16	Arm akimbo and turn the head sideways
	B8	D3, D15, D16, D17	Arm akimbo and point by hand
E3 (Sadness)	B9	D1, D7, D8, D11, D12	Crouch and shake the head
	B10	D2, D3, D4, D15, D16, D17	Keep hands in front of body
	B11	D3, D4, D6, D13, D15, D16	Bend forward and hands down
E4 (Surprise)	B12	D3, D4, D15, D16	Raise arms vertically
	B13	D6, D7, D12, D13	Bend backward
E5 (Fear)	B14	D3, D4, D6, D7, D8, D11, D12, D13, D15, D16	Bend backward, hold the head in hands, crouch down
	B15	D6, D7, D8, D9, D10, D11, D12, D13	Stand back
	B16	D3, D4, D7, D8, D11, D12, D15, D16	Crouch down and hold the head in hands
E6 (Disgust)	B17	D15, D16, D17	Push one arm out
	B18	D3, D4, D15, D16, D17	Take back one arm and the other arm down
	B19	D3, D4, D15, D16	Clamp the arms

* The behaviors are shown in Figure 2, and the definitions of different DoFs are illustrated in Figure 1.

**Table 2 biomimetics-09-00684-t002:** Four intensity levels and corresponding recognition rates of adjusted emotional categories.

Intensity Level	E1 * (Rate)	E2 * (Rate)	E3 * (Rate)	E4 * (Rate)	E5 * (Rate)	E6 * (Rate)
L1	B4 (95.45%)	-	-	-	B16 (83.33%)	-
L2	B1 (90.91%)	-	-	-	B15 (81.82%) B14 (68.18%)	-
L3	B12 (63.64%) B5 (57.57%)	B7 (65.15%)	B9 (56.06%) B14 (37.88%) B11 (66.67%)	B12 (36.36%) B13 (48.48%)	B9 (51.51%)	B17 (63.64%)
L4	B2 (51.52%)	B8 (56.06%) B6 (39.39%)	B16 (18.18%)	-	B19 (46.97%)	B15 (18.18%) B7 (31.82%)

* E1~E6 represent happiness, anger, sadness, surprise, fear, and disgust, respectively.

## Data Availability

The original contributions presented in the study are included in the article. Further inquiries can be directed to the corresponding author.

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
