# Peer review of "Human Perception of the Emotional Expressions of Humanoid Robot Body Movements: Evidence from Survey and Eye-Tracking Measurements"

_biomimetics, 2024, doi:10.3390/biomimetics9110684_

Round 1
Reviewer 1 Report
Comments and Suggestions for Authors
Thank you for submitting this interesting paper. I have no substantive critical comments, it is a clear and well-written paper with a need for minor English editing.
Comments on the Quality of English LanguageMinor edits.
Reviewer 2 Report
Comments and Suggestions for Authors
The paper is interesting and can be accepted.
Reviewer 3 Report
Comments and Suggestions for Authors
The manuscript “Human’s Perception towards the Emotional Expressions of Humanoid Robot’s Body Movements: Evidence from Survey and Eye-tracking Measures” has many important qualities. The manuscript is well-written and the findings make a valuable contribution to the field. Additionally, the authors do a commendable job of addressing most of the study's limitations in a well-structured Discussion section, which is greatly appreciated. Overall, it is a solid piece of research. However, there are a few aspects that need to be addressed before it can be considered for publication. I am confident that the authors can easily resolve these minor issues. Please see the points listed below.
Lines 176 & 177. Since six specific emotions were analyzed, the term "including" should be removed. It implies the presence of additional emotions, which is not the case here. Although English is not my first language, I am quite certain that "including" is not appropriate in this context.
Table 1. Tables and figures should be independent from the main text. Please, at least inform the meaning of the emotion codes. For example: E1 = happiness. See journal instructions for the right place where this and other information should be placed.
Line 187. Avoid using “etc.” in methods. Please put all the information here or direct the reader to where he/she can find all the details.
Line 190. Please inform the mean and standard or error deviation for the age range.
Line 191. Change to: “in which 33 were males and 33 were females”. That is, the word “including” is misplaced here.
Line 199. "Including" means that other devices were used. If this is the case, inform all devices used in the study. Otherwise, suppress the word "including". Also, inform the company that produces the devices, if applicable.
Line 208. Please inform future readers how the random order was achieved. For example, was a program used for this purpose?
Figure 5. Please inform future readers about the error bars in the legend. Do they represent the standard deviations?
Line 255. It would be helpful for readers to include a brief explanation about how pupil diameter serves as an indicator of emotions. This could be added where the authors initially mention using pupil changes to gauge emotional states.
Figure 6. See the comment about Figure 5.
Lines 429-436. Many studies point to cultural differences involving body postures. I suggest adding this aspect to the study's limitations.
Reviewer 4 Report
Comments and Suggestions for Authors
Minor comments:
1) Figures 5, 6, and 7 are difficult to see (they are too small).
2) Perhaps the movement of pinching the nose should correspond to disgust.
Major questions:
3) Regarding the bodily expression of emotion, the authors should cite Darwin's book "The Expression of the Emotions in Man and Animals."
4) The authors should justify why they used different numbers of behaviors for each emotion (e.g., B1-B5 for happiness and B12-B13 for surprise).
5) 66 or 69 participants? Both numbers appear on page 5.
6) An even number of emotion intensities (L1-l4) is not the best choice. It should be an odd number so that the middle value corresponds to an average intensity.
7) Figure 4 and Table 2 should be better explained. For instance, in Table 2, shouldn't the percentages for each emotion add up to 100%? Why was the average value 3.5 chosen as the threshold of intensity?
8) The authors assume that the reactions measured in the human participants were caused by the robots. I'm not sure to what extent this is 100% accurate.
Round 2
Reviewer 4 Report
Comments and Suggestions for Authors
No further comments.